# Insight into the Amelioration Effect of Nitric Acid-Modified Biochar on Saline Soil Physicochemical Properties and Plant Growth

**DOI:** 10.3390/plants13233434

**Published:** 2024-12-06

**Authors:** Lei Yan, Guang Gao, Mu Lu, Muhammad Riaz, Mengyang Zhang, Kaiqing Tong, Hualong Yu, Yu Yang, Wenjing Hao, Yusheng Niu

**Affiliations:** 1Institute of Biomedical Engineering, College of Life Sciences, Qingdao University, Qingdao 266071, China; yanlei2022@qdu.edu.cn (L.Y.); gaoguang178@163.com (G.G.); lumu1218@163.com (M.L.); cordelia30@163.com (K.T.); yuhualong2022@163.com (H.Y.); yangyu34@qdu.edu.cn (Y.Y.); haowenjing@qdu.edu.cn (W.H.); 2College of Resources and Environment, Zhongkai University of Agriculture and Engineering, Guangzhou 510225, China; riaz1480@hotmail.com; 3College of Resources and Environment, Qingdao Agricultural University, Qingdao 266109, China; zhangmengyang@qau.edu.cn; 4Academy of Dongying Efficient Agricultural Technology and Industry on Saline and Alkaline Land in Collaboration with Qingdao Agricultural University, Dongying 257347, China; 5National Center of Technology Innovation for Comprehensive Utilization of Saline-Alkai Land, Dongying 257347, China

**Keywords:** salinized soil, biochar modification, characterization, soil fertility, oxidative damage, pakchoi

## Abstract

Soil salinization is a major factor threatening global food security. Soil improvement strategies are therefore of great importance in mitigating the adverse effect of salt stress. Our study aimed to evaluate the effect of biochar (BC) and nitric acid-modified biochar (HBC) (1%, 2%, and 3%; m/m) on the properties of salinized soils and the morphological and physiological characteristics of pakchoi. Compared with BC, HBC exhibited a lower pH and released more alkaline elements, reflected in reduced contents of K^+^, Ca^2+^, and Mg^2+^, while its hydrophilicity and polarity increased. Additionally, the microporous structure of HBC was altered, showing a rougher surface, larger pore size, pore volume, specific surface area, and carboxyl and aliphatic carbon content, along with lower aromatic carbon content and crystallinity. Moreover, HBC application abated the pH of saline soil. Both BC and HBC treatments decreased the sodium absorption rate (SAR) of saline soil as their concentration increased. Conversely, both types of biochar enhanced the cation exchange capacity (CEC), organic matter, alkali-hydrolyzable nitrogen, and available phosphorus and potassium content in saline soils, with HBC demonstrating a more potent improvement effect. Furthermore, biochar application promoted the growth-related parameters in pakchoi, and reduced proline and Na^+^ content, whilst increasing leaf K^+^ content under salt stress. Biochar also enhanced the activity of key antioxidant enzymes (superoxide dismutase (SOD), peroxidase (POD), and catalase (CAT)) in leaves, and reduced hydrogen peroxide (H_2_O_2_) and malondialdehyde (MDA) content. Collectively, modified biochar can enhance soil quality and promote plant growth in saline soils.

## 1. Introduction

The exacerbation of factors such as improper irrigation, environmental degradation, and global climate change has led to increased soil salinity, which in turn damages plant growth and productivity [1]. High salinity adversely affects approximately 20% of arable land [2,3]. Additionally, salt stress can also lead to an excessive accumulation of reactive oxygen species (ROS), hindered protein synthesis, and distorted protein structures, reducing plant growth and crop yield [4,5,6]. By 2050, food production must increase by 70%, requiring a 50% rise in grain output to 3 billion tons [7], while about 50% of arable land will be affected by salinity [8]. Therefore, innovative technologies to treat salt-stressed soils are urgently needed to address the growing food crisis. Nowadays, enhancing saline soils and crop growth have become key research priorities to meet food demand and ensure sustainable agriculture. Current approaches focus on three primary strategies: (1) desalinating seawater through underground drainage and other engineering methods; (2) improving nutrient utilization in saline soils through agricultural practices such as straw returning and fertilization; and (3) optimizing soil conditions with chemical amendments like soil conditioners. To mitigate salt stress, plants upregulate protective mechanisms, including antioxidant-based detoxification (both enzymatic and non-enzymatic), secondary metabolite accumulation, and ion redistribution [9,10]. The use of soil amendments like biochar to improve saline soils is an increasingly promising technology.

Biochar is an efficient, environmentally friendly, carbon-rich, and fine-grained porous material produced through the high-temperature pyrolysis of biomass waste (e.g., straw, wood, poultry manure) in anaerobic conditions [11]. Biochar pH generally increases with pyrolysis temperature, making it typically alkaline [12]. With a high specific surface area and oxygen-containing functional groups, biochar exhibits excellent adsorption properties, hydrophilicity or hydrophobicity, and acid-base buffering capacity [13,14]. The biomass source used in pyrolysis also influences biochar structure, specific surface area, pore volume, and pore size distribution [15,16]. These properties make biochar a commonly used amendment for acidic soils. Biochar application can improve soil water retention, enhance organic matter and organic carbon levels, increase concentrations of soil nutrients (such as NO_3_^−^-N, available P, K^+^, Ca^2+^, Mg^2+^), boost aggregate stability, and promote nutrient cycling. Recently, modified biochar has been increasingly applied to improve saline–alkali soils [17,18], proving that biochar can reduce pH and salinity in coastal saline soils [19,20]. Biochar has improved soil structure and micro-properties, enhanced water-holding capacity, total organic carbon, available K, and CEC, and significantly boosted wheat [21,22] and corn [23] growth in salt-affected soils. The nutrients released by biochar also contribute to improved microbial community structure and enzyme activity in soil [21,24]. Biochar application alleviates salt stress in rice by modifying soil properties and regulating soil microbial abundance and community composition [25].

In addition to soil improvements, biochar influences plant physiological responses and enhances salt tolerance. Biochar plays a key role in reducing Na⁺ content in plants due to its rich pore structure, which lowers Na^+^ availability in the soil and releases essential elements like Ca^2+^, K^+^, and Mg^2+^ [26]. In saline soils, biochar helps maintain a favorable Na^+^/K^+^ balance in plants, reduces oxidative stress, and thereby promotes plant growth and biomass [27,28]. Biochar-enhanced soil properties also increase photosynthetic activity, chlorophyll synthesis, gene expression, stress-responsive protein activity, and maintain osmotic enzyme and hormone balance [29]. Proline, produced by a protected response by biochar, aids in detoxifying ROS in plant cells. Biochar also promotes the production of osmotic substances like proline and soluble polysaccharides, which regulate osmotic balance and reduce Na^+^ accumulation [30,31]. Biochar supports plant root and shoots growth by reducing Na⁺ levels, indirectly lowering endogenous stress hormones (such as abscisic acid and polyamine oxidase) and growth hormone (IAA) levels [28].

Limited studies have explored the characteristics of nitric acid-modified biochar (HNO₃-modified biochar, HBC) and its role in triggering stress-responsive mechanisms in pakchoi seedlings, including soil physical and chemical characteristics and other critical plant stress responses in saline soils. Therefore, this study aimed to (i) investigate the differences in the physicochemical properties of biochar before and after modification using advanced characterization techniques such as Fourier-transform infrared spectroscopy (FTIR), X-ray diffraction (XRD), and solid-state nuclear magnetic resonance (^13^C-NMR); (ii) examine the effects of HBC on alleviating soil salt damage and improving soil nutrient levels, and (iii) assess the potential of HBC to enhance the growth potential and physiological characteristics of salt-stressed pakchoi.

## 2. Materials and Methods

### 2.1. Preparation of Modified Biochar

Primitive biochar (BC) was a pyrolysis product of maize straw provided by Shenyang Agricultural University, China. It was separated from oxygen and cracked at 500 °C for 2 h, and then was ground and passed through a 0.25 mm sieve for later application. The 0.1 M HNO_3_-modified biochar (HBC) was prepared by the impregnation method (Appendix A). Briefly, 10 g of BC was added to 100 mL of 0.1 M HNO_3_ solution and stirred the suspension at 200 rpm for 4 h at 25 °C [32]. The acidified biochar was filtered from the solution, and then thoroughly washed with deionized water until the pH remained constant. Finally, the obtained modified biochar was dried in a blast oven at 60 °C to obtain HBC, which was collected for use.

### 2.2. Characterization of Modified Biochar

A 2.50 g biochar sample was placed in 50 mL of CO_2_-free ultra-pure water, boiled for 5 min, and replenished with evaporated water. The sample was filtered, with the initial 5 mL of filtrate discarded. After cooling the remaining liquid to room temperature, the pH was measured using a pH meter. The elemental composition (C, N, H, S, and O) of the biochar was analyzed with an elemental analyzer (VarioElcube, Germany) [33], and the primary characteristics of BC and HBC are presented in Table 1.

The biochar-specific surface area was measured using the Brunauer–Emmett–Teller (BET) method, while pore volume and pore size were calculated using the Barrett–Joyner–Halenda (BJH) method. The morphology and surface structure of the biochar samples were examined by scanning electron microscopy (SEM), and the elemental distribution (C, N, O, Ca, Mg, Al, Fe, and Si) on the surface was analyzed with scanning electron microscopy–energy-dispersive spectroscopy (SEM-EDS), with parameters set as follows: acceleration voltage at 20 kV, working distance at 7.3 mm, and 9000× magnification. In addition, ^13^C nuclear magnetic resonance (^13^C-NMR, Bruker Biospin, Karlsruhe, Germany) spectra were used to analyze the various carbon structures in the biochar. Surface functional groups and crystal minerals in the biochar samples were identified using Fourier-transform infrared spectroscopy (FTIR) (Thermo, Waltham, MA, USA) and X-ray diffraction (Bruker D8 ADVANCE, Billerica, MA, USA), respectively.

### 2.3. Experimental Design and Soil Properties

The saline soil used in the experiment was collected from Binzhou, Shandong Province (118°02′ E, 37°22′ N). After air drying, the soil was sieved through a 2 mm mesh to remove plant debris. The soil physical and chemical properties were as follows: pH 7.88, salinity 5.11‰, organic matter 1.618 g/kg, alkali-hydro N 35.178 mg/kg, available P 11.523 mg/kg, and available K 211.3 mg/kg.

Indoor soil incubation experiments were conducted without fertilization. Seven treatments were set up: CK (without biochar), BC1 (1% BC, *w*/*w*), BC2 (2% BC, *w*/*w*), BC3 (3% BC, *w*/*w*), HBC1 (1% HBC, *w*/*w*), HBC2 (2% HBC, *w*/*w*), and HBC3 (3% HBC, *w*/*w*). BC and HBC were mixed evenly with saline soil and placed in 50 mL centrifuge tubes. The tubes were incubated in a plant cultivation chamber at the School of Life Sciences, Qingdao University, China. Each treatment included three replicates, arranged in a completely randomized design. Water was applied daily to maintain appropriate soil moisture content. The incubation lasted for three months.

### 2.4. Soil Physicochemical Analysis

After the biochar incubation period, the soil was air-dried to a constant weight and sieved through 10, 20, and 100 mesh screens for further analysis. The physicochemical properties of the soil were analyzed using standard methods [34]. Soil pH was measured in a 1:2.5 soil-to-water suspension (*w*/*v*) using a digital pH meter (FE20, Shanghai Mettler Toledo Co., Shanghai, China). Soil organic matter content was determined by the K_2_Cr_2_O_7_-H_2_SO_4_ volumetric method. Alkali-hydrolyzable nitrogen content was measured by the alkaline hydrolysis diffusion method. Available phosphorus and potassium were extracted with NaHCO_3_ and CH_3_COONH_4_, respectively, and analyzed by ion chromatography.

Soil cation exchange capacity (CEC) was determined using 1.75 g of air-dried soil passed through a 10-mesh sieve, placed in a 50 mL centrifuge tube with 25.0 mL of 1.66 cmol/L [Co(NH_3_)_6_]Cl_3_ solution, and incubated at 20 °C for 60 min. After centrifuging at 4000 rpm for 10 min, the absorbance was measured at 475 nm. The concentrations of water-soluble Na^+^, Mg^2+^, and Ca^2+^ in the soil were separately measured, and the sodium absorption rate (SAR) was calculated using the following formula:$$SAR=\frac{Na^{+}}{\sqrt{\left(Ca^{2+}+Mg^{2+}\right)/2}}$$

### 2.5. Plant Potting Experiment

A pot experiment was conducted with a 2% biochar. The soil used in the experiment was collected from Qingdao University, Shandong Province (120°43′ E, 36°07′ N). The soil’s physical and chemical properties were as follows: pH 6.67, organic matter 9.32 g/kg, alkali-hydro N 118 mg/kg, available P 37.7 mg/kg, and available K 223.9 mg/kg. Three treatments were set up: CK (without biochar), BC (2% BC), and HBC (2% HBC). Each pot was filled with 1.0 kg of air-dried soil sieved through a 2 mm mesh. Fertilizers were applied at rates of 0.3 g/kg soil for urea (N), 0.1 g/kg soil for Na_2_HPO_4_·12H_2_O (P_2_O_5_), and 0.3 g/kg soil for KCl (K_2_O), with an additional 3‰ NaCl. Pakchoi seeds (*Brassica campestris* L. ssp. Chinensis) were soaked for 4–6 h for germination, then sown (10 seeds per pot) and thinned to three plants per pot post-germination. Soil moisture was maintained at 60% of the maximum water-holding capacity, which was adjusted daily. The experiment was conducted in an incubator under a 16/8 h day/night cycle at 25 °C. Each treatment was repeated three times in a completely randomized design.

### 2.6. Plant Seedling Growth Potential, and Measurement of Na^+^ and K^+^ in Leaf

After 15 days of treatment, the pakchoi seedlings were harvested, carefully removed from the pots, and rinsed with deionized water. Plant height, root length, and fresh weight were measured using a graduated ruler and a precision scale.

A 0.2 g dry sample of ground plant material was dry-ashed at 500 °C in a muffle furnace for 4 h, dissolved in 10 mL of 0.1 M HCl, and Na^+^ and K^+^ concentrations were determined by ion chromatography (Essentia IC-16i, Shimadzu, Kyoto, Japan).

### 2.7. Determination of Proline and Malondialdehyde (MDA) Content

A 0.2 g fresh sample was extracted with 5 mL of 3% sulfosalicylic acid solution, frozen, and ground. A 2 mL aliquot of the extraction solution, 2 mL glacial acetic acid, and 2 mL of acidic ninhydrin reagent were heated in a boiling water bath for 30 min. After cooling, 4 mL toluene was added, and proline content in the upper solution was measured at 520 nm [35].

For MDA content, the thiobarbituric acid (TBA) method was used according to the Vos et al. method [36]. The optical density (OD) value of the supernatant was measured at wavelengths of 532 nm, 600 nm, and 450 nm, respectively. The MDA content is expressed as µmol/g fresh sample (μmol/g FW).

### 2.8. Analysis of Antioxidant Enzyme Activity of SOD, POD, and CAT

A 0.5 g fresh samples were homogenized at 4 °C in 0.05 M phosphate-buffer solution (PBS, pH 7.8) and centrifuged at 15,000× *g* for 15 min. The supernatant was used to determine enzyme activity.

Peroxidase (POD) activity was measured using the guaiacol method [37], which was measured by H_2_O_2_ degradation at 470 nm spectrophotometry for 3 min and expressed as [(u·(g·min)^−1^]. The activity of superoxide dismutase (SOD) was defined as the unit required to inhibit NBT photoreduction by 50% [38]. Catalase (CAT) activity was determined by observing the disappearance of H_2_O_2_ absorbance at 240 nm as the method of Aebi [39].

### 2.9. Statistical Analysis

Data were analyzed by one-way ANOVA using Duncan’s test (*p* < 0.05) based on SPSS software. FTIR spectroscopy data were processed using OMNIC 8.2 (Thermo Fisher Scientific, Waltham, MA, USA). All charts were drawn using Origin Professional 8.6 (Origin Lab Corporation, Northampton, MA, USA) software.

## 3. Results

### 3.1. Analysis of Microstructure and Surface Components of BC and HBC

SEM results showed that the surface of BC had a smooth structure with similarly sized micropores arranged in a regular pattern (Figure 1A,C). In contrast, HBC exhibited a rougher surface, a damaged carbon framework, and pores of uneven sizes, with some pores elongated and enlarged. The pore structure of HBC appeared cleaner (Figure 1B,D). EDS results revealed that the surface C content of HBC decreased by 3.29% (from 80.06% to 77.43%) compared to BC, while O content increased by 36.66% (from 13.53% to 18.49%), with a specific increase in Fe content. Furthermore, the content of alkaline elements such as K, Ca, and Mg on the HBC surface decreased by 90.27%, 74.15%, and 21.43%, respectively, compared to BC (Figure 1E,F).

The specific surface area, pore volume, and pore size of biochar analyzed using BET and BJH methods, significantly increased after HNO_3_ modification, with HBC reaching 66.3699 m^2^/g compared to BC 25.6082 m^2^/g. Similarly, the average pore diameter and size of HBC increased by 26.94- and 43.48-fold compared to BC (Figure 1G–I; Appendix A).

The nitrogen adsorption–desorption isotherms at 77K indicate that both BC and HBC exhibited type IV isotherm characteristics, with a desorption hysteresis loop suggesting stronger adsorption capacity in HBC than BC. The modified biochar had a broad pore size distribution and increased pore volume, primarily in the 2.0–5.0 nm range (Figure 1H,I).

### 3.2. Analysis of Elements in BC and HBC

After modification with HNO_3_, the C and H content in biochar decreased, while the N and O content increased by 18.85-fold and 1.21-fold, respectively. The H/C, O/C, and (O + N)/C ratios in HBC increased relative to BC, with H/C increasing by 7.14%, O/C by 47.5%, and (O + N)/C by 47.62% (Table 1).

### 3.3. XRD and FTIR Analysis in BC and HBC

The crystal phase of biochar remains unchanged before and after modification, with similar peak patterns. Three characteristic peaks were observed: 2θ = 20.88° (identified as KAlCl_3_O_8_), 2θ = 26.6° (identified as Al_2_SiO_5_), and 2θ = 28.3° (identified as CaCO_3_). Among them, the relative intensity of these peaks changed, with an increase at 2θ = 20.88° and 2θ = 26.6° in HBC and a decrease at 2θ = 20.83° compared to BC (Figure 2A,B).

The FTIR spectra of BC and HBC exhibited similar characteristic peaks at 3433 cm^−1^ (-OH), 1590 cm^−1^ (C=C), and 1040 cm^−1^ (Si-O-Si), with higher intensities in HBC. Notably, HBC shows additional peaks at 1137 cm^−1^ and 1187 cm^−1^, corresponding to C-O-C vibration (Figure 2C,D).

### 3.4. ^13^C-NMR Analysis of Organic Carbon Structures in BC and HBC

BC and HBC carbon structures primarily consist of alkyl carbon (0–50 ppm), alkoxy carbon (50–100 ppm), aromatic carbon (100–160 ppm), and carbonyl/carboxyl/ester carbon (160–200 ppm).

Compared with the original biochar, the peaks of the spectrum of biochar modified with nitric acid show significant differences between the aliphatic carbon region at 0–100 ppm and the carboxyl carbon region at 160–220 ppm (Figure 3). Specifically, compared to BC, HBC decreased the signals of the CH_3_, O-CH_3_, C-C, and C-H groups, whilst increasing the signals of the C-O, C-N, and COO groups (Table 2).

### 3.5. The Effect of Biochar on pH, SAR, and CEC in Salinized Soil

Compared to the control, BC treatments at 1%, 2%, and 3% showed no significant effect on soil pH, while HBC at the same doses resulted in minor pH reductions, particularly HBC2, which decreased pH by 1.13% (from 7.90 to 7.81) (Figure 4A).

Indeed, the SAR in BC1, BC2, and BC3 was gradually lower than that in CK; similarly, HBC1, HBC2, and HBC3 exhibited the same trend, and there was no significant difference between BC and HBC at the same dose (Figure 4B). The soil treated with BC2 and BC3 exhibited enhanced CEC when compared to biochar-untreated treatment, increased by 60.79% and 87.21%; HBC noticeably enhances soil CEC, and the degree of increase is higher than that of the BC (Figure 4C).

### 3.6. The Effect of Biochar on Physico-Chemical Properties in Salinized Soil

Salt stress adversely impacted soil properties. Figure 5 shows that compared to the treatment without BC (CK), adding biochar significantly improved salt-affected soil organic matter and alkali-hydro N content as biochar concentration increased. While BC and HBC at 1% showed no significant differences, HBC at 2% and 3% had higher values (Figure 5A,B).

In comparison to the CK, available P and K contents also increased with biochar addition, with the highest increases observed in HBC-treated soils, particularly at the 2% concentration (Figure 5C,D).

### 3.7. The Effect of Biochar on Plant Growth Potential and the Leaf Na^+^ and K^+^ Content in Salinized Soil

Biochar has a beneficial effect on plant growth and physiological characteristics, and it appears that HBC treatment is more effective (Figure 6A). Specifically, the plant height moderately increased under BC2 and HBC2 treatments, although there was no significant effect between the two treatments (Figure 6B). BC2 and HBC2 treatments sequentially increased the root length and fresh weight of whole seedlings (Figure 6B,C).

Biochar noticeably reduced Na^+^ content in the leaf in comparison to the absence of biochar, with a reduction of 16.13% (from 10.29 to 8.63 mg g^−1^ DW) and 30.42% (from 10.29 to 7.16 mg g^−1^ DW) in BC2 and HBC2-treated leaves, respectively (Figure 6D). An opposite trend was found in the leaf K^+^ content, enhanced by 1.73- and 1.78-fold in BC2 and HBC2 treatments, respectively (Figure 6E). The figure below shows whether or not modified biochar (BC2 and HBC2) exposure resulted in the decrease in Na^+^/K^+^ ratio in the leaf with or without modified biochar, and the reduction is more significant with HBC2 (Figure 6F).

### 3.8. The Effect of Biochar on Antioxidant Enzyme in Salinized Soil

POD activity increased in both BC2 and HBC2 treatments relative to the control, although the difference between the two biochar treatments was not significant (Appendix A). The activities of CAT and SOD were remarkably increased under BC2 and HBC2 treatments, and the HBC2 exhibited higher activity, with increases of 1.43- and 1.91-fold, respectively (Appendix A).

### 3.9. The Effect of Biochar on Proline and MDA Content in Salinized Soil

The proline content was drastically inhibited after employing biochar, whether modified or not, and HBC2 exhibited the lowest proline content, decreased by 13.59% (from 83.84 to 72.45 µg g^−1^ FW) and 18.54% (from 83.84 to 68.30 µg g^−1^ FW), respectively (Figure 7A). Similarly, compared to the CK, BC2 and HBC2 treatments significantly reduced MDA accumulation, with BC showing a more pronounced effect (Figure 7B).

## 4. Discussion

### 4.1. Acid Modified Biochar Exhibites Better Biological Properties

Improving saline soil and developing novel salt-tolerant genotypes are essential mechanisms in managing salt-affected agricultural production [40]. In our study, we found that HNO_3_ immersion reduced the pH of biochar (Table 1). The elemental composition of biochar can predict its polarity, degree of carbonization, and hydrophobicity, with the H/C and O/C molar ratios typically used to estimate these properties [41]. Herein, HNO_3_-modified biochar (HBC) showed changes in its C, O, and H content (Table 1), indicating transformation in its functional groups, as confirmed by FTIR results. Specifically, the relative absorption peaks of HBC at 1100 cm^−1^, 1600 cm^−1^, and 3440 cm^−1^ shifted compared to untreated biochar (BC) (Figure 2C,D). The increase in O content and decrease in H content in HBC (Table 1) may result from (i) the oxidation of hydroxyl groups in cellulose by HNO_3_, and (ii) HNO_3_ oxidizing properties, which break the aromatic structure and introduce O-containing functional groups, such as carboxyl groups, increasing biochar hydrophilicity and enhancing its cation absorption [42]. ^13^C-NMR spectroscopy analysis also indicated an increase in O-containing functional groups (Figure 3; Table 2), consistent with the findings of Liu et al. [43].

The modification effects of oxidants like HNO_3_ can disrupt the original surface structure of biochar, making its pore structure more developed [44]. Studies have shown that a larger surface area and a developed pore structure in biochar significantly improve soil physical properties [45]. SEM results revealed a rough surface in HNO_3_-modified biochar, with varied pore sizes and some elongated or expanded pore channels (Figure 1A–D), likely due to HNO_3_ dissolving ash materials. The acid modification increases the BET-specific surface area, while alkali modification shows the opposite effect [46]. BET data indicated that the specific surface area of HBC is significantly larger than that of BC (Figure 1G–I; Appendix A), likely due to low-concentration HNO_3_ forming a loose and porous carbon structure. Furthermore, strong HNO_3_ oxidation partially damages the surface and skeleton of biochar, opening previously closed pores, leading to higher average pore size and volume in HBC compared to BC (Figure 1G–I), aligning with the studies of Wan et al. [47] and Li et al. [48]. N_2_ adsorption and desorption experiments further confirmed a predominantly mesoporous structure in biochar. The physical adsorption results also indicate a higher Na^+^ adsorption capacity in HBC, potentially reducing Na^+^ absorption by crops. Generally, material crystallinity is inversely related to surface defects [49]. XRD results showed a decrease in HBC crystallinity, suggesting that modification disrupts the crystal structure and alters biochar stability (Figure 2), potentially enhancing pore structure and increasing biochar adsorption capacity [50].

### 4.2. Biochar Improves Soil Fertility and Enhances Plant Resistance

Biochar, typically an alkaline material, can increase soil pH, leading to extensive research on its use in improving acidic soils [51]. Recent studies suggest that acid-modified biochar also improves saline soils. Our study showed that acidified biochar reduces soil pH (Figure 4A). The sodium adsorption ratio (SAR), representing the relative abundance of Na^+^, Ca^2+^, and Mg^2+^ in soil solution, decreased with the addition of BC and HBC (Figure 4B), which is consistent with the findings that biochar effectively improves salt-affected soil quality and reduces Na^+^ toxicity in plants [52]. Both BC and HBC increased soil CEC (Figure 4C), likely due to the (i) Ca^2^⁺ from biochar displacing Na^+^ on soil colloids or, due to its high specific surface area and CEC, adsorbing more Na^+^, thereby reducing SAR; and (ii) biochar’s porous structure promoting soil aggregation and permeability [52], facilitating Ca^2+^, Mg^2+^, and Na^+^ displacement and accelerating Na^+^ leaching [53,54].

Biochar can reduce soil nitrogen (N) leaching by adsorbing NO_3_^−^-N through pores, chelation, and electrostatic interactions [55]. Acid modification introduces acidic functional groups on the biochar’s surface, enhancing nitrogen absorption [56]. H_2_SO_4_-modified biochar, for instance, shows improved ammonia N adsorption compared to unmodified biochar [57]. Biochar, as an organic matter source, significantly increases soil organic matter [58], a finding consistent with our results showing enhanced organic matter in salt-affected soil post treatment with BC and HBC, particularly at 2% and 3% HBC (Figure 5A). Other studies have demonstrated that biochar increases soil N, P, and K content [59,60] by limiting N immobilization, increasing net ammonification and nitrification, and retaining inorganic N in soil [61,62]. Our results show that HBC increases N content in salt-stressed soil (Figure 5B). Biochar also retains NH_4_^+^-N through acidic functional groups, countering N loss in saline soil [63,64] and increasing available K and P content [19], which aligns with our findings that HBC-2 treatment had the highest available P and K levels (Figure 5C,D), similar to observations found in the Yellow River Delta [65].

### 4.3. Biochar Regulates Plant Antioxidant Defense and Ion Homeostasis, Enhancing Plant Salt Resistance

To further explore biochar’s effects on saline soil, we analyzed pakchoi growth using 2% biochar. Biochar improved the rhizosphere environment in saline soils, reducing salt-induced damage and promoting plant growth. Biochar enhances soil moisture retention and reduces Na^+^ concentration in root zones, promoting plant health [66]. Biochar mitigates salt-stress-induced oxidative damage, as seen in barley [67], wheat [22], sorghum [68], and corn [23]. In our study, both biochar types improved pakchoi growth under salt stress, with modified biochar showing more significant effects on root length and fresh weight (Figure 6A–C), indicating that biochar can indeed alleviate salt alkali stress and promote plant growth, and the effect of improved biochar is more obvious [69]. Salt stress induces an excessive production of ROS, which has devastating effects on plant membranes, proteins, and lipids [4,5]. The antioxidant enzyme defense system plays a crucial role in protecting plants from oxidative damage caused by salt stress [70,71]. Biochar reduced plant H_2_O_2_ levels under salt stress, with further reductions after adding modified biochar (Figure 7B), and increased the activity of antioxidant enzymes’ POD, CAT, and SOD (Figure 6D–F), suggesting that biochar alleviates ROS damage by enhancing the antioxidant defense system, leading to reduced membrane permeability and lower MDA content (Figure 7C). Proline functions as an osmo-protectant and ROS scavenger [72]. Biochar reduced proline accumulation in salt-affected soil, especially with modified biochar (Figure 7A). Under salt stress, biochar application reduced Na^+^ and increased K^+^ content in leaves, lowering the Na^+^/K^+^ ratio. Similar effects in *Sesbania cannabina* [65] support biochar’s role in enhancing plant salt tolerance by maintaining ion balance and reducing salt-stress resistance. This study assumes that the effects of biochar and HNO_3_-modified biochar (HBC) are consistent across different saline soils and plant species, which may not account for variability in soil types, climates, or crops. The short-term experiments may not reflect long-term stability, and interactions with soil microbes or organic matter were not considered. Future work should explore the long-term stability of biochar in diverse soils, its interaction with soil microbial communities, and its effects on a wider range of crops under varying environmental conditions.

## 5. Conclusions

HNO_3_ modification effectively enhances biochar’s ability to improve saline soil. HBC application lowers soil pH and Na^+^ adsorption rate, with its modified surface structure and functional groups promoting nutrient adsorption, reducing soil alkali-hydro N, available P, and K loss, and increasing organic matter and CEC in salt-affected soil. Notably, a 2% HBC showed the most improvement in soil fertility, indicating that HNO_3_-modified biochar is effective for soil improvement. Additionally, biochar applications reduced salt-stress-induced oxidative damage by enhancing antioxidant defenses and reducing osmotic stress through a decreased proline and Na^+^/K^+^ ratio in leaves, with modified biochar offering enhanced benefits.

## Figures and Tables

**Figure 1 plants-13-03434-f001:**
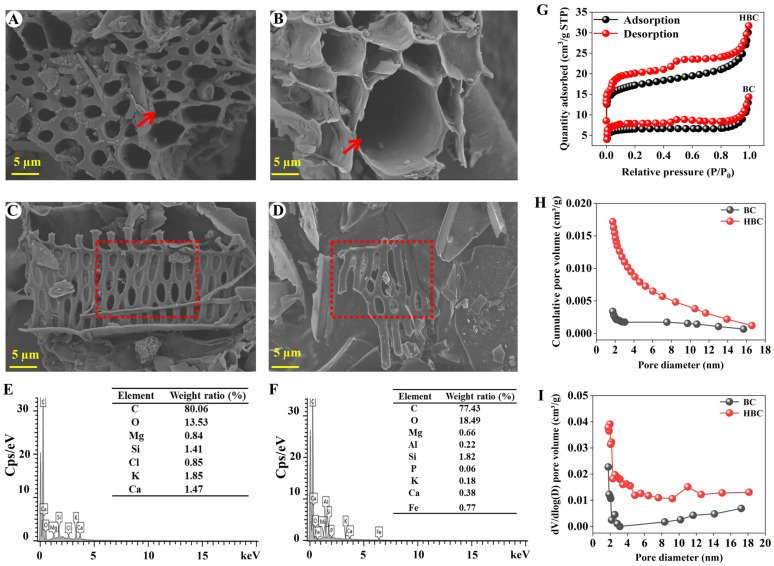
The surface characteristics of biochar samples. (**A**,**C**): Scanning electron microscopy (SEM) images of BC (primitive biochar); (**B**,**D**): SEM images of HBC (HNO_3_-modified biochar); (**E**,**F**): EDS of BC and HBC; (**G**): N_2_ isotherm adsorption–desorption curves; (**H**): cumulative pore volume of different sizes; (**I**): pore size distribution. The red arrow indicates the pore size in the biochar; The red box represents the surface damage of biochar.

**Figure 2 plants-13-03434-f002:**
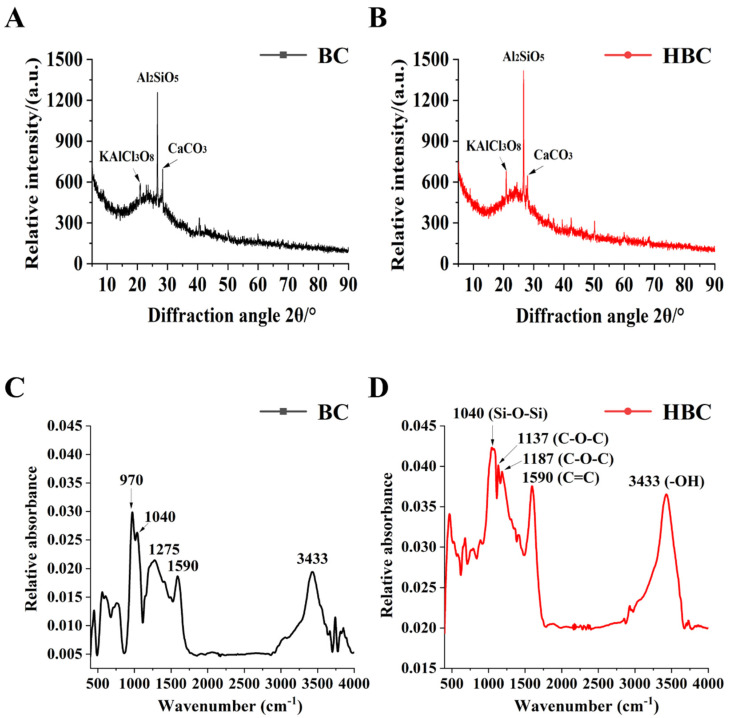
The X-ray diffraction (XRD) (**A**,**B**) and Fourier transform infrared spectroscopy (FTIR) (**C**,**D**) characteristic analysis performed on BC (**A**,**C**) and HBC (**B**,**D**). The relative content of surface functional groups in BC and HBC was reflected in FTIR spectra (500−4000 cm^−1^).

**Figure 3 plants-13-03434-f003:**
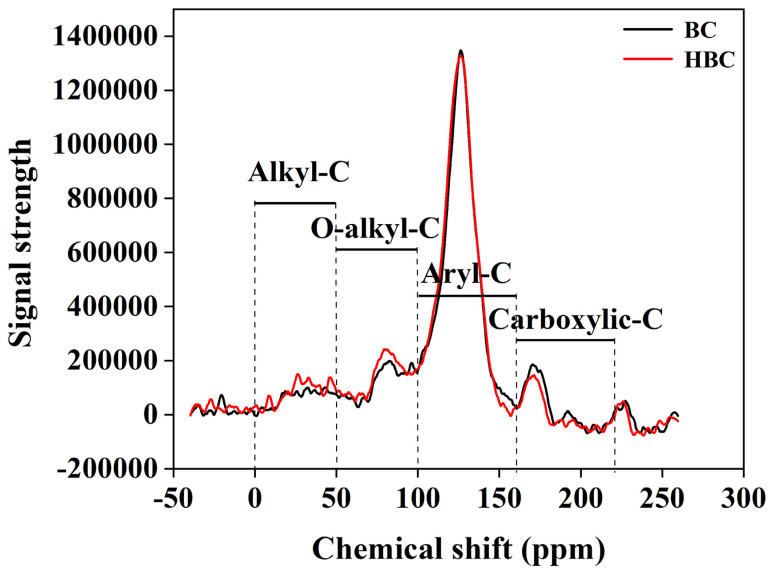
^13^C nuclear magnetic resonance (^13^C-NMR) spectra of BC and HBC.

**Figure 4 plants-13-03434-f004:**
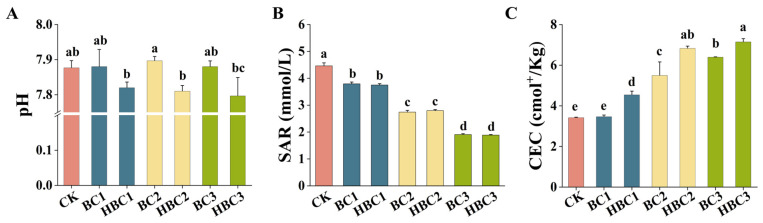
The effects of different addition ratios of BC and HBC on the pH (**A**), SAR (**B**), and CEC (**C**) of salinized soil. Plants treated without biochar (CK); plants treated with 1% BC (BC1); plants treated with 2% BC (BC2); plants treated with 3% BC (BC3); plants treated with 1% HBC (HBC1); plants treated with 2% HBC (HBC2); plants treated with 3% HBC (HBC3). Different letters (a–e) represent significant differences at a 95% probability level.

**Figure 5 plants-13-03434-f005:**
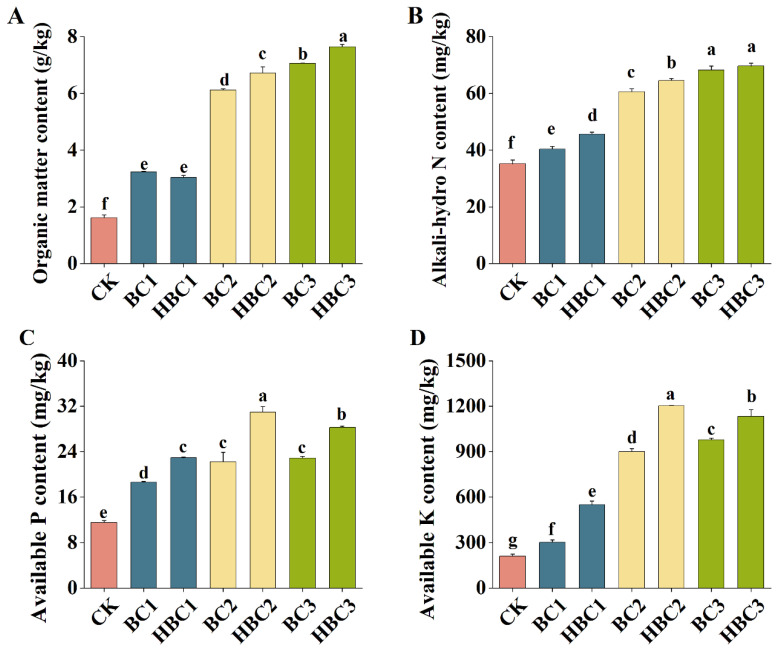
The effects of different addition ratios of BC and HBC on the physico-chemical properties in salinized soil. (**A**): organic matter content; (**B**): alkali-hydro N content; (**C**): available P content; (**D**): available K content. Different letters (a–g) represent significant differences at a 95% probability level.

**Figure 6 plants-13-03434-f006:**
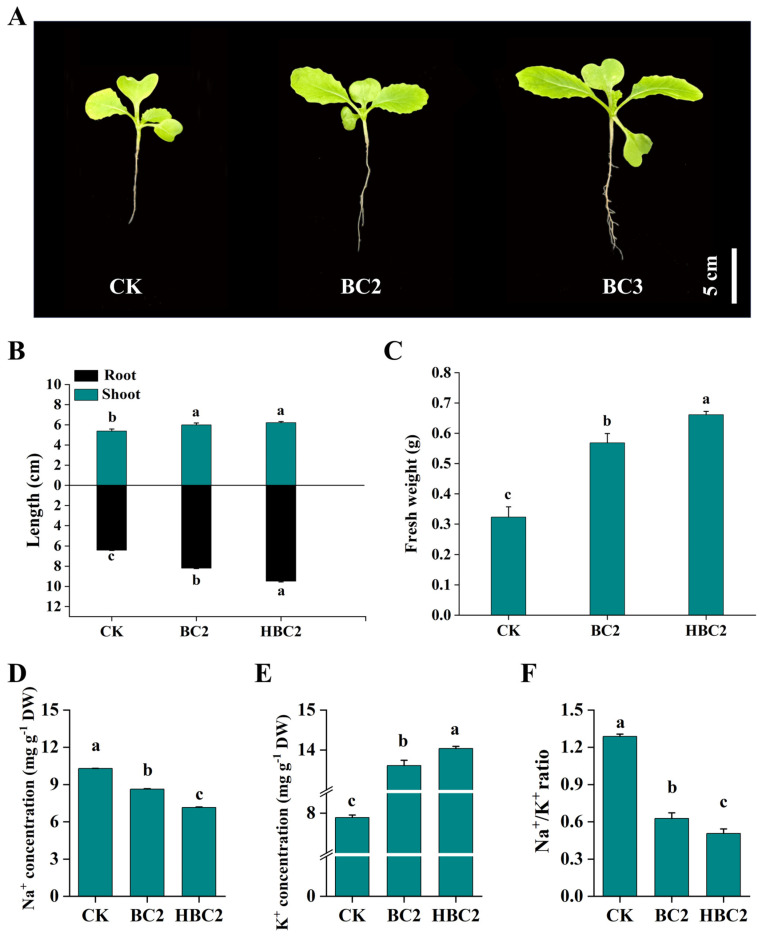
The effects of BC and HBC on the growth potential and ion content of pakchoi seedlings. (**A**): the growing tendency of seedlings; (**B**): root length and shoot height; (**C**): fresh weight of whole seedlings; (**D**): the content of Na^+^ in leaves; (**E**): the content of K^+^ in leaves; (**F**): the Na^+^/K^+^ ratio in leaves. Plants treated without biochar (CK); plants treated with 2% BC (BC2); plants treated with 2% HBC (HBC2). Different letters (a–c) represent significant differences at a 95% probability level.

**Figure 7 plants-13-03434-f007:**
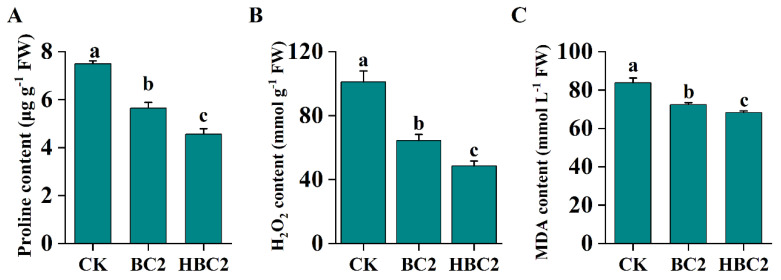
The effects of BC and HBC on the content of proline (**A**), H_2_O_2_ (**B**) and MDA (**C**) of the leaves of pakchoi seedlings. Different letters (a–c) represent significant differences at a 95% probability level.

**Table 1 plants-13-03434-t001:** Basic properties of BC and HBC samples.

	C (%)	N (%)	H (%)	S (%)	O (%)	H/C	(O + N)/C	O/C	C/N	pH
BC	59.69	1.30	1.95	0.46	24.51	0.03	0.43	0.41	45.92	9.85
HBC	50.40	1.67	1.49	0.42	29.73	0.03	0.62	0.59	30.18	5.59

Note: H/C, O/C, and (O + N)/C reflect the aromaticity, hydrophilicity, and polarity of biochar samples.

**Table 2 plants-13-03434-t002:** The distribution of carbon functional groups and the relative content of different organic carbons (%) in BC and HBC.

	Alkyl-C	O-Alkyl-C	Aryl-C		Carboxylic-C	
	0–50 ppm	50–100 ppm	100–140 ppm	140–160 ppm	160–190 ppm	190–220 ppm
	CH_3_	O-CH_3_	C-C, C-H	C-O, C-N	COOH, COOC	COC
BC	0.106	0.176	0.696	0.056	0.036	-
HBC	0.074	0.144	0.654	0.064	0.054	-

## Data Availability

All relevant data supporting our findings are available in the manuscript or from the corresponding author upon request.

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
