# Peer review of "Insight into the Amelioration Effect of Nitric Acid-Modified Biochar on Saline Soil Physicochemical Properties and Plant Growth"

_plants, 2024, doi:10.3390/plants13233434_

Round 1
Reviewer 1 Report
Comments and Suggestions for Authors
This manuscript deals with the effect of modified biochar on saline soil properties and plant growth. The study is really interesting and very well structured and documented, providing new insights into the potential use of modified biochar as a mean to address soil salinity and improve plant response. Especially, the sections of introduction and discussion are adequately documented with enriched literature at the subject of the study. Therefore, I would suggest it for publication.
Few corrections
-Figures 1 and 4 should be magnified.
Authors should add a paragraph to discuss the bias and assumptions from the methods used in the study. Perhaps before conclusion.
Future work should also be provided.
Author Response
Comments and Suggestions for Authors
This manuscript deals with the effect of modified biochar on saline soil properties and plant growth. The study is really interesting and very well structured and documented, providing new insights into the potential use of modified biochar as a mean to address soil salinity and improve plant response. Especially, the sections of introduction and discussion are adequately documented with enriched literature at the subject of the study. Therefore, I would suggest it for publication.
Few corrections
- -Figures 1 and 4 should be magnified.
Response:
Thank you so much for raising this valuable comment. The modifications have been completed as required. we hope the revised manuscript will satisfy the editor and reviewers.
Figure 1. The surface characters of biochar samples.
Figure 4. Effects of different addition ratios of BC and HBC on the pH (A), SAR (B), and CEC (C) of salinized soil.
- Authors should add a paragraph to discuss the bias and assumptions from the methods used in the study. Perhaps before conclusion.
Response:
Many thanks. The suggestions/comments raised by the worthy reviewer in the report file, have been incorporated. We have added a short statement at the mentioned place.
“This study assumes that the effects of biochar and HNO3-modified biochar (HBC) are consistent across different saline soils and plant species, which may not account for variability in soil types, climates, or crops. The short-term experiments may not reflect long-term stability, and interactions with soil microbes or organic matter were not considered. Future work should explore the long-term stability of biochar in diverse soils, its interaction with soil microbial communities, and its effects on a wider range of crops under varying environmental conditions.”
- Future work should also be provided.
Response:
Thank you for your observation and suggestion. We have added it in the resubmitted manuscript.

Reviewer 2 Report
Comments and Suggestions for Authors
I congratulate you on your study, but to improve the version prepared for publication, I have the following suggestions:
Title
The title is appropriate for the research topic.
Abstract
The abstract is well-written.
Aim of the Study
The aim of the study is clearly and correctly stated.
Introduction
The introduction is well-structured; however, I suggest that the transition between ideas be made smoother (the connection between the discussion on salinity and the use of biochar could be better emphasized).
Materials and Methods
The materials and methods are appropriate for the study and are well-described. To enhance this section, I recommend adding a brief justification for the HNO3 concentration used.
Results
The study provides a comprehensive analysis of the differences between untreated biochar (BC) and nitric acid-treated biochar (HBC), highlighting the improvements achieved through chemical modification. The results are well-structured and cover essential aspects such as physicochemical properties, impacts on saline soils, and effects on plant growth.
Discussion and Conclusions
This section is well-structured, offering an excellent integration of experimental results with the existing literature.
References
There are self-citations, but they align with the topic of the current study.
Author Response
Comments and Suggestions for Authors
- I congratulate you on your study, but to improve the version prepared for publication, I have the following suggestions:
Title
The title is appropriate for the research topic.
Abstract
The abstract is well-written.
Aim of the Study
The aim of the study is clearly and correctly stated.
Response:
Thank you for your recognition.
- Introduction
The introduction is well-structured; however, I suggest that the transition between ideas be made smoother (the connection between the discussion on salinity and the use of biochar could be better emphasized).
Response:
Many thanks, we have reformulated and merged the paragraph. Therefore, this smooth transition between salinity and biochar.
- Materials and Methods
The materials and methods are appropriate for the study and are well-described. To enhance this section, I recommend adding a brief justification for the HNO3 concentration used.
Response:
Many thanks for your nice comment. According to your good suggestion, we have added brief discussion as HNO3-modified biochar (HBC) was prepared by the impregnation method (Fig. S1). Briefly, add 10 g of BC to 100 mL of 0.1 M HNO3 solution, and stir the suspension at 200 rpm for 4 h at 25 ℃
All the changes are indicated with blue text in the manuscript, and we hope the revised manuscript will satisfy the editor and reviewers.
- Results
The study provides a comprehensive analysis of the differences between untreated biochar (BC) and nitric acid-treated biochar (HBC), highlighting the improvements achieved through chemical modification. The results are well-structured and cover essential aspects such as physicochemical properties, impacts on saline soils, and effects on plant growth.
Discussion and Conclusions
This section is well-structured, offering an excellent integration of experimental results with the existing literature.
Response:
Thank you.
- References
There are self-citations, but they align with the topic of the current study.
Response:
Thank you for supportive comments, we always follow the journal guidelines.

Reviewer 3 Report
Comments and Suggestions for Authors
In this study aimed to evaluate the effects of biochar (BC) and nitric-acid modified biochar (HBC) on the properties of salved soil and the morphology and physiological characteristics of pakichoi. compare to BC and HBC exhibited a lower pH and released more alkaline elements, reflected in reduced content of K+, Ca+2 and Mg+2. Generaly biochar reduce the sodium Na+ and increace K+ and reduce plant stress.
1. What soil and from what place wascollected for experiments using classic and modified biochar? On the sodium and potassium concentration chart, please use the same scale up to 15.
Author Response
Comments and Suggestions for Authors
In this study aimed to evaluate the effects of biochar (BC) and nitric-acid modified biochar (HBC) on the properties of salved soil and the morphology and physiological characteristics of pakichoi. compare to BC and HBC exhibited a lower pH and released more alkaline elements, reflected in reduced content of K+, Ca2+ and Mg2+. Generally, biochar reduce the sodium Na+ and increase K+ and reduce plant stress.
- What soil and from what place was collected for experiments using classic and modified biochar?
Response:
Many thanks for your nice comments/suggestions. We have added this information as the saline soil used in the experiment was collected from Binzhou, Shandong Province (118°02'E, 37°22′N).
- On the sodium and potassium concentration chart, please use the same scale up to 15
Response:
Thank you, it has been done in the revised manuscript.
Figure 6. Effects of BC and HBC on the growth potential and ion content of pakchoi seedlings.
